# Assessment of the Risk of Crack Formation at a Hybrid Bonding Interface Using Numerical Analysis

**DOI:** 10.3390/mi15111332

**Published:** 2024-10-30

**Authors:** Xuan-Bach Le, Sung-Hoon Choa

**Affiliations:** 1Graduate School of Nano IT Design Fusion, Seoul National University of Science and Technology, Seoul 01811, Republic of Korea; bach.lx.hust@gmail.com; 2Intelligent Semiconductor Engineering Department, Seoul National University of Science and Technology, Seoul 01811, Republic of Korea

**Keywords:** hybrid bonding, crack formation, peeling stress, chemical mechanical polishing (CMP) defect, finite element analysis (FEA)

## Abstract

Hybrid bonding technology has recently emerged as a promising solution for advanced semiconductor packaging technologies. However, several reliability issues still pose challenges for commercialization. In this study, we investigated the possibility of crack formation caused by chemical mechanical polishing (CMP) defects and the misalignment of the hybrid bonding structure. Crack formation and thermomechanical stress were analyzed for two common hybrid bonding structures with misalignment using a numerical simulation. The effects of annealing temperature and dishing value on changes in the non-bonding area and peeling stress were systematically analyzed. The calculated peeling stresses were compared to the bonding strength of each bonding interface to find vulnerable regions prone to cracking. The non-bonding area in the bonding structure increased with a decreasing annealing temperature and an increasing dishing value. To achieve a sufficient bonding area of more than 90%, the annealing temperature should be greater than 200 °C. During the heating period of the annealing process, the SiCN-to-SiCN bonding interface was the most vulnerable cracking site with the highest peeling stress. An annealing temperature of 350 °C carries a significant risk of cracking. On the other hand, an annealing temperature lower than 250 °C will minimize the chance of cracking. The SiCN-to-SiO_2_ bonding interface, which has the lowest adhesion energy and a large coefficient of thermal expansion (CTE) mismatch, was expected to be another possible cracking site. During cooling, the SiCN-to-Cu bonding interface was the most vulnerable site with the highest stress. However, the simulated peeling stresses were lower than the adhesion strength of the bonded interface, indicating that the chance of cracking during the cooling process was very low. This study provides insights into minimizing the non-bonding area and preventing crack formation, thereby enhancing the reliability of hybrid bonding structures.

## 1. Introduction

These days, the semiconductor industry requires fine-pitch interconnections and high-performance semiconductor devices. Therefore, advanced semiconductor packaging has emerged as the next frontier in the development of high-performance devices, with the aim of achieving higher density within a smaller footprint. Advances in high-speed computing, autonomous driving, and artificial intelligence (AI) have driven the development of advanced packaging technologies such as fan-out wafer-level packaging (FOWLP), chip-on-wafer-on-substrate (CoWoS) packaging, and 3D packaging [1,2].

Among advanced packaging technologies, hybrid bonding technology has emerged as a promising solution for achieving high performance and fine pitch interconnection. Hybrid bonding is a direct bonding method that combines metallic and dielectric materials to form robust interconnections. Copper (Cu) is typically used as a metallic material, while materials such as SiO_2_, SiCN, or polyimide are commonly used as dielectric materials [3,4,5]. Unlike traditional bonding methods, which have minimum interconnect pitches of 20–50 µm [6], hybrid bonding can achieve pitches of less than 1 µm, making it advantageous for 3D integrated circuits (3D ICs) and heterogeneous integration [7,8,9].

Despite its advantages, hybrid bonding technology accompanies numerous challenges. One of the critical issues is bonding misalignment [10,11]. Bonding misalignment can result in various reliability issues, including the reduction in the Cu-to-Cu bonding area, leakage current, and copper diffusion in the bonding structure. Another critical issue is the thermomechanical stress caused by the mismatch of the coefficient of thermal expansion (CTE) among the materials used in hybrid bonding. Delamination or cracking at bonding interfaces can occur due to high peeling stress during the annealing process. Lin et al. [12,13] reported that the annealing temperature and dishing value played crucial roles in the Cu bonding area’s formation and peeling stress. A high annealing temperature or a low dishing value may cause dielectric interface cracking due to high peeling stress during the heating period. Conversely, low annealing temperature or a high dishing value can reduce the Cu bonding area or even cause void formation due to insufficient Cu protrusion during the heating period. However, the crack formation mechanism in hybrid bonding structures during the annealing process has not yet been thoroughly studied. 

Performing a chemical mechanical polishing (CMP) process before the bonding process is crucial to ensure a high level of surface roughness and flatness, which are essential process conditions for the success of hybrid bonding [14,15,16]. However, the CMP process can lead to potential CMP defects that affect the quality of the hybrid bonding interface. Fujino et al. [17] reported that excessive corrosion can occur in regions with multiple metal layers such as a Cu/barrier layer due to galvanic corrosion during the CMP process. Due to the galvanic corrosion effect, voids were observed at the edge of the Cu pad after the annealing process [18].

In this study, we investigated the effects of misalignment and CMP defects on the integrity of hybrid bonding using a numerical analysis. We analyzed the thermomechanical stress of two common Cu pad structures with misalignment for different annealing temperatures and dishing depths and identified vulnerable regions prone to cracking. Additionally, we investigated the possibility of non-bonding region and crack formation during the annealing process due to CMP defects caused by galvanic corrosion. The obtained results will provide in-depth insights into the non-bonding area and crack formation mechanisms during the annealing process and offer detailed guidelines to prevent the thermomechanical cracking of hybrid bonding structures.

## 2. Theoretical Background

### 2.1. CMP Defects in Hybrid Bonding

The basic process sequence of hybrid bonding is shown in Figure 1. The process begins with the deposition of SiO_2_ and SiCN layers onto a silicon wafer through plasma enhanced chemical vapor deposition (PECVD) (step a). Next, dielectric etching is performed to create cavities (step b). The barrier layers of Ta/TaN are then deposited (step c). After barrier deposition, Cu electroplating is used to fill the etched cavities, forming Cu pads (step d). Then, chemical mechanical polishing (CMP) is applied to remove excess Cu and ensure precise control of the Cu dishing value. Once CMP is completed, the dry plasma activation between the top and bottom wafer is applied to create reactive dangling bonds on the dielectric surface (step f). These dangling bonds react with hydrogen in the air to form –OH groups, resulting in van der Waals bonding at room temperature between two dielectric surfaces (step g). Finally, a post-bonding annealing process is conducted to create strong Cu bonds due to the protrusion and diffusion of Cu [19,20].

The successful implementation of hybrid bonding strongly depends on the CMP process. The CMP process involves the use of CMP slurries containing abrasive particles and reactive chemicals to planarize the wafer surface. Typically, the CMP process includes two steps: Cu CMP to remove excessive Cu and barrier CMP to eliminate the barrier layer. During the CMP process, several different chemical reactions can occur. Among these, galvanic corrosion on regions with many metal layers such as the Cu/barrier (Cu/Ta/TaN) interface is particularly significant. This corrosion, illustrated in Figure 2a, occurs due to the CMP slurry providing an ionic contact environment due to the difference in the electrochemical potential between Cu and the barrier layers. As a result, the less noble metal (Cu) acting as the anode will corrode at a higher rate than the more noble metal (barrier) acting as the cathode [16,21]. This leads to excessive corrosion at the interface between the Cu and barrier during the CMP process. This corrosion is considered the cause of voids observed after the annealing process, as illustrated in Figure 2b [17,18,22,23].

The post-bonding annealing process includes a heating period and a cooling period. During the heating period, since Cu has a higher CTE than the dielectric materials, the Cu pads expand more than the surrounding dielectric materials. Consequently, the gap between the Cu pads on the top and bottom wafers gradually closes due to a Cu protrusion effect. The Cu protrusion during the heating period includes both elastic deformation and plastic deformation. As the temperature increases, the Cu protrusion also increases until the Cu pads on the top and bottom wafers come into contact each other. Once the Cu pads on the top and bottom wafers contact each other, the compressive stress is generated at the Cu bonding interface as Cu protrusion continues during the heating period. The repulsive force generated by the Cu pads when they come into contact with each other can create undesirable high tensile stress on the surrounding dielectric interface (SiCN-to-SiCN and SiCN-to-SiO_2_ interfaces). Consequently, the dielectric bonding interface may undergo debonding if the tensile stress exceeds the dielectric bond strength. The tensile stress in the direction perpendicular to the bonding interfaces is commonly referred to as peeling stress (σ_yy_) [12,24]. The maximum Cu protrusion occurs at the end of the heating period of annealing, just before the cooling period begins. The peeling stress on the dielectric bonding interface thus reaches its highest value at the end of the heating period. During the cooling period, the Cu pads on the top and bottom wafers will shrink, and the elastically deformed Cu pad will return to its original shape. As a result, the Cu bonding interface is now subjected to peeling stress (σ_yy_), while the dielectric bonding interface is subjected to compressive stress [13,25].

### 2.2. Numerical Modeling

Hybrid bonding takes place based on the connection between the Cu pads and surrounding dielectrics of the top and bottom wafers. In reality, there will always be misalignment between the Cu pads of the top wafer and the Cu pads of the bottom wafer [4,12,17,26,27]. This misalignment depends on the accuracy of the alignment equipment. When misalignment occurs, several reliability issues are expected, such as the reduction in the Cu-to-Cu bonding area and a buildup of thermomechanical stress in the bonding structure, leading to an increasing risk of crack generation in the hybrid bonding structure. Therefore, in this study, we focused on the two most common structures in hybrid bonding. Figure 3a shows one of the common hybrid bonding structures with different Cu pad sizes, where the bottom Cu pad in the bottom wafer is larger than the top Cu pad in the top wafer. This bonding structure is denoted as the type A structure in this study. Even though this structure has no bonding misalignment, the Cu is in direct contact with the dielectric materials, and thus the bonded interface between the Cu pad and dielectric materials and CMP defects at the edge of the Cu pad is susceptible to peeling-off and cracking.

In this study, we used two different dielectric materials, SiO_2_ and SiCN, which are commonly used in hybrid bonding technology. SiCN dielectric materials have recently been used due to the advantage of a higher bonding energy than SiO_2_ at a low annealing temperature. Additionally, SiCN is known to be a good barrier to Cu diffusion and electromigration [28,29]. However, fabricating SiCN can be expensive, and SiCN can increase the warpage of the structure. For these reasons, multilayer SiCN deposited on SiO_2_ is widely used in industries [18,28,29,30]. The thicknesses of the SiCN and SiO_2_ layers in this study were 0.5 μm and 1.7 μm. In the type A structure, the top Cu pad had a diameter of 3 μm, while the bottom Cu pad had a diameter of 5 μm. Figure 3b shows a magnified view of the edge of the hybrid bonding structure, which is the critical region with interfaces, where cracks often appear during the annealing process [12,17,27]. Therefore, in this study, we focused on this critical region.

The type B structure, as shown in Figure 3c, represents the typical hybrid bonding structure having bonding misalignment. The type B structure had the same Cu pad size of 4 μm for both wafers, with 1 μm misalignment. Both type A and type B structures had Cu pad thicknesses of 2 μm. The Ta/TaN barrier layer was modeled with a total thickness of 0.01 μm, and the wafer thickness was 20 μm in this study.

To investigate the effects of bonding conditions on crack generation in hybrid bonding, we used a local model for a single hybrid bonding structure in the wafer. This approach is recognized as an effective way to save computational efforts while maintaining numerical accuracy [31,32]. Table 1 shows the dimensions of the hybrid bonding structure for the local model used in this study, which is commonly used in fine-pitch semiconductor packaging [33,34].

A 2D plane strain FEA model was developed using ANSYS Workbench 2022 software. The corresponding mesh plot for the type A structure is shown in Figure 4a. To set up the boundary conditions of a local model of the hybrid bonding structure, we used symmetry conditions on the left edge of the model and a specific condition on the right edge (U_Y_ = free, U_X_ = free, and U_X_ had the same value for all nodes on the right edge). The PLANE183 element was chosen for analysis with a total of 73,658 elements. The FEA model for the type B structure was similar to the type A structure, with only the adjustment of the dimension and position of the Cu pads. 

The meshing at the Cu pad corners, where CMP defects occurred, is shown in Figure 4b. The dishing depth caused by galvanic corrosion at the Cu/barrier region was assumed to be 5 nm since it is known to be typically 5 nm larger than the average dishing value for the CMP process [12,35,36]. The initial dishing profile with the actual surface curvature and surface roughness was simplified and modeled as a flat surface. This was because the dishing depth is typically in a range of only a few nanometers, which is very small compared to the Cu pad dimension of several micrometers. Hence, the dishing profile was modeled as being flat in the FEA model.

The annealing temperature profile is plotted in Figure 5. The annealing process was divided into a heating period (including a ramp-up phase and a dwell phase) and a cooling period (a cool-down phase). Table 2 lists the process parameters used in this study.

The following assumptions were established in the FEA model:Share nodes were used at the boundaries of the Cu/barrier layer surfaces.The SiCN-to-SiCN bonds formed before the post-bonding annealing process were considered ‘bonded type’ contact.For the Cu-to-Cu interface and SiCN-to-Cu interface, ‘rough type’ contact was applied during the heating period. ‘Bonded type’ contact was applied during the cooling period with a contact radius of 4 Å. In other words, during the heating period, the distance between the Cu (or SiCN) of the top wafer and the Cu (or SiCN) of the bottom wafer smaller than 4 Å was considered to be ‘bonded type’ contact during the cooling period.

The elastic material properties used in the numerical analysis are presented in Table 3. The stress-free state was considered at room temperature. In addition to material elasticity, the plasticity and creep effects of Cu played a significant role during the annealing process due to the high annealing temperature and dwell time. High annealing temperatures cause irreversible Cu deformation due to plasticity and creep effects. Therefore, the plastic and creep material properties for Cu were applied, and a bilinear plastic model of Cu was used with a corresponding yield strength of 321 MPa and a tangent modulus of 2000 MPa [12,37]. 

As for the Cu creep model, the time-dependent creep characteristics of Cu were used. Specifically, we used the general creep relationship (given in the equation below) with the creep strain rate ($\dot{\varepsilon }$), applied stress (σ), temperature (T), and time (t). This model was suitable for Cu materials subjected to high temperature and fatigue loading conditions.
$$\dot{\varepsilon_{cr}}=A\exp\left(-\frac{Q}{RT}\right)\sigma^{n}t^{m}$$

The material constants for Cu were A = 1.43 × 10^10^, the activation energy Q = 197 kJ mol^−1^, the universal gas constant R = 8.314 J mol^−1^K^−1^, n = 2.5, and m = −0.9 [12,37,38,39].

## 3. Results and Discussion

### 3.1. Analysis of Deformation Behavior and Non-Bonding Area

In this section, we investigate the deformation behavior and change in the bonding area during the annealing process, which are important for the integrity of the bonding structure. In this study, the bonding interface in the hybrid bonding structure includes the Cu-to-Cu, SiCN-to-SiCN, and SiCN-to-Cu interfaces. The overall bonding strength of the hybrid interface is determined by the cumulative strengths of these three different bonding interfaces at each contact region. It is evident that an increase in the bonding area will enhance the overall bonding strength. Meanwhile, the formation of voids at the bonding interface will reduce the overall bonding strength. 

Before we conduct a detailed numerical analysis, we will review the bonding strength at the hybrid bonding interfaces, which are Cu-to-Cu, SiCN-to-SiCN, SiCN-to-SiO_2_, and SiCN-to-Cu, from previous papers. The bonding or adhesion strength values depend on the material properties and annealing temperatures. The Cu-to-Cu diffusion bonding shows very high bonding energy, which can reach up to 12 J/m^2^ at 300 °C annealing [44] and potentially increase to 25–40 J/m^2^ [45]. The bonding energy of SiCN-to-SiCN ranges from 1.4 J/m^2^ to 2.5 J/m^2^ for low annealing temperatures of 200–250 °C and is as high as 3 J/m^2^ at 300 °C annealing [28,29,46]. Meanwhile, the bonding energy of SiCN-to-SiO_2_ is relatively low at 1.6 J/m^2^ [47]. On the other hand, the Cu-SiCN bonding energy in misalignment regions relies heavily on plasma surface treatment. In previous reports, the bonding energy of Cu-SiCN was found to be 1.2–5.5 J/m^2^ using fracture mechanics tests [45,48,49]. Therefore, it is thought that the weakest bonding interface in this study is the SiCN-to-SiO_2_ interface, which has the lowest bonding energy, followed by the SiCN-to-SiCN and SiCN-to-Cu interfaces. The Cu-to-Cu interface is the strongest bonding interface.

Figure 6a shows the total deformation map of the bonding structure after the annealing process for the type A structure at 300 °C annealing with 5 nm dishing. It is observed that after the annealing process, two void regions appear at the bonding interfaces: region I and region II (the right corners of the Cu pad). The voids form because the Cu protrusion during the annealing process is not sufficient to cover the entire dishing volume, especially the CMP defects. Figure 6b provides a magnified view of the deformation map at region I, where Cu-to-Cu and SiCN-to-Cu bonding are formed. Figure 6c presents a magnified view of the deformation map at region II, where SiCN-to-Cu bonding is formed. In Figure 6b,c, the most vulnerable points for potential cracking are point 1 and point 2, which are the bonding interface at the edges of voids. In particular, in region II (Figure 6c), point 1 is the boundary between the bonding/non-bonding area of the SiCN-to-Cu bonding interface, whereas point 2 is the boundary between the non-bonding/bonding area of the SiCN-to-SiCN bonding interface. These points are usually subjected to high stress concentrations and act as crack tips for crack initiation. The void length directly affects the bonding area, leading to the reduced bonding strength and reliability of the hybrid bonding, and can even result in the failure of the hybrid bonding. Therefore, the non-bonding area and the risk of crack generation at point 1 and point 2 should be thoroughly investigated. 

Figure 7 presents the effects of the annealing temperature and dishing value on the changes in the non-bonding area for different annealing temperatures from 200 °C to 350 °C. Here, the non-bonding area represents the void size, as shown in Figure 6b,c. It is observed that the non-bonding area increases with an increasing dishing value and lower annealing temperature. The Cu protrusion increases with an increasing annealing temperature. Therefore, lower annealing temperatures result in reduced Cu protrusion and consequently larger non-bonding areas. Meanwhile, a larger dishing value requires more Cu protrusion to cover the gaps, which also leads to a larger non-bonding area. 

Figure 7a shows the effect of annealing temperatures and dishing values on the non-bonding area in region I of the type A structure. The non-bonding area increases rapidly at 200 °C annealing due to insufficient Cu protrusion. Therefore, the dishing value has a significant impact on the non-bonding area at the 200 °C annealing condition. The non-bonding area will be more than 10% of the total area if the dishing value becomes more than 5 nm. On the other hand, when the annealing temperature is 250 °C, the non-bonding area is less than 10% (which means that the bonding area is more than 90%) for all dishing values. At 300 °C annealing, the non-bonding area is significantly reduced and was less than 5%. When the annealing temperature increases to 350 °C, the non-bonding area becomes very small (less than 2%) regardless of the dishing values (corresponding to a bonding area greater than 98%). 

Figure 7b shows the effect of the annealing temperature and dishing value on the non-bonding area in region II for the type A structure shown in Figure 6c. The non-bonding area in region II tends to be larger than in region I in Figure 6b. The non-bonding area in region II also increases rapidly at 200 °C annealing. This is due to the deformation of the Cu pad into a convex shape during Cu protrusion, with the Cu protrusion at region II (the outermost edge of the Cu pad) being lower than that at region I [23,50]. This difference becomes significant at low annealing temperatures, combined with a large dishing value. Specifically, at 200 °C annealing, even with a 3 nm dishing value, the non-bonding area reaches 13%. The non-bonding area further decreases when the annealing temperature is increased to 300 °C or higher. 

Figure 7c,d illustrate the effects of annealing temperatures and dishing values on the non-bonding areas of the type B structure for the region I and region II bonding interface, respectively. The trends of the non-bonding area in the type B structure are similar to those in the type A structure. However, the non-bonding area of the type B structure tends to be larger than that of the type A structure. This difference becomes significant at lower annealing temperatures and larger dishing values. These results are attributed to the smaller volume of the bottom Cu pad of the type B structure with smaller Cu protrusion during annealing relative to that of the type A structure, resulting in an increased non-bonding area. 

In this study, we can use a threshold of a 10% non-bonding area (corresponding to a 90% bonding area) as a guideline to ensure the high quality of bonding without the delamination of the dielectric materials, as also suggested in several previous studies [12,25,51]. In summary, in order to achieve a non-bonding area of less than 10% (corresponding to a bonding area of more than 90%), the annealing temperature should be greater than 200 °C, which is consistent with results reported in previous experimental studies [12,33,34]. When an annealing temperature of 250 °C is used, the dishing value should be less than 5 nm. When the annealing temperature reaches 300 °C or higher, the non-bonding area will be less than 10% for all dishing values. On the other hand, the type A structure is advantageous for reducing the non-bonding area compared to the type B structure. 

### 3.2. Analysis of the Effect of Bonding Conditions on Crack Formation

#### 3.2.1. Effect of Bonding Conditions During the Heating Period

The generation of cracks due to high stress is another critical factor that can lead to the fracture or failure of the hybrid bonding structure. Therefore, we focused on assessing the risk of crack formation at the bonding interfaces during the heating and cooling period of the annealing process. An analysis of the overall stress in the hybrid bonding structure and high stress concentration is crucial and requires detailed investigation. In this section, we focus on identifying weak regions prone to crack generation. The primary stress components responsible for crack generation at the bonding interface are analyzed. 

Figure 8a displays the distributions of von Mises stress at the end of the heating period for the type A structure, which is subjected to a 300 °C annealing process with a 5 nm dishing value. The highest von Mises stress occurs at point 2 in region II (the SiCN-to-SiCN bonding interface), reaching a value of 794 MPa, indicating that the point 2 location in region II is the most critical site. To further investigate the primary driving force for crack generation at the bonding interface, the distribution of peeling stress (σ_yy_) at region II is analyzed and the results are shown in Figure 8b. The highest peeling stress also occurs at point 2, reaching 680 MPa.

Figure 9a exhibits the von Mises stress and peeling stress distribution for the type B structure (at 300 °C annealing and 5 nm dishing) at the end of the heating period, respectively. Similar to the type A structure, the type B structure shows the highest von Mises stress and peeling stress at point 2 in region II, reaching values of 675 MPa and 458 MPa, respectively. The stress level of the type B structure is lower than that of the type A structure, which is attributed to the lower volume of the bottom Cu pad compared to that of the type A structure. During annealing, the larger Cu pad expands more and exerts larger stress on adjacent dielectric materials. 

Previous studies have shown that bulk SiCN has a tensile strength of 1200 MPa and a critical energy release rate (G_C_) of 12.2 J/m^2^ [43,52]. Additionally, the average adhesion energy of the SiCN-to-SiCN bonding interface is roughly 2.5 J/m^2^. By using the following equation, the critical peeling stress (σ_C_) for the SiCN-to-SiCN interface can be roughly estimated [53,54]:$$\sigma_{C}=\sqrt{\frac{G_{C}\times E}{Y^{2}\times a\times \left(1-{ʋ}^{2}\right)}}$$
where σ_C_ is critical peeling stress, G_C_ is adhesion energy, E is Young’s modulus, Y is crack shape factor, a is crack length, and ʋ is Poisson’s ratio. 

Based on the above equation, the critical peeling stress of the SiCN-to-SiCN interface is estimated to be around 550 MPa [42,43,52]. Comparing the critical peeling stress with the calculated peeling stress of the type A structure (680 MPa) and type B structure (458 MPa) at 300 °C annealing, it can be postulated that the type B structure has almost no risk of cracking, whereas the type A structure is likely to experience cracking at the SiCN-to-SiCN bonding interface during the annealing process. Also, it can be noted that the actual adhesion strength at the corners of the non-bonding area may be lower than the average adhesion strength of the SiCN-to-SiCN bonding interface, which may further increase the risk of crack formation. 

From the stress analysis for both type A and type B structures during the heating period, point 2 in region II is identified as the initiation point for crack propagation. The peeling stress induces Mode I crack propagation at the SiCN-to-SiCN bonding interface at point 2. This phenomenon has also been reported in experiments conducted by Lin et al. and Miao et al. [12,24], where cracks were observed at the dielectric bonding interface at the point 2 location. Another possible cracking site would be the SiCN-to-SiO_2_ bonding interface, which has the lowest adhesion energy in the hybrid bonding structure. Also, there was significant CTE mismatch between the SiCN layer and the SiO_2_ layer. Thus, the SiCN-to-SiO_2_ bonding interface is another vulnerable site for cracking. In particular, if cracks at point 2 propagate vertically downwards (thickness direction), the propagated cracks will readily promote the cracking of the SiCN-to-SiO_2_ bonding interface. 

Next, we investigate the effects of the annealing temperature on the peeling stress. The effects of annealing temperature on the peeling stress distribution in region II are illustrated in Figure 10 for the type A structure and Figure 11 for the type B structure. It is evident that increasing the annealing temperature will increase the peeling stress. This can be attributed to the increased Cu protrusion resulting in an increase in the pushing force exerted by the Cu pads on the surrounding SiCN materials, thereby increasing the peeling stress of the dielectric SiCN layer. 

In the case of annealing temperatures of 200 °C and 250 °C, the peeling stress is low for both the type A and type B structures. Specifically, the peeling stress at point 2 with annealing temperatures of 200 °C and 250 °C (in Figure 10a,b) is less than 50 MPa, indicating a very low risk of crack generation. When the annealing temperature increases to 300 °C, the peeling stress begins to increase sharply for both the type A and type B structures. The peeling stress increases more significantly when the annealing temperature reaches 350 °C. Specifically, for the type A structure, as shown in Figure 10c, the peeling stress at point 2 increases significantly up to 1340 MPa when annealed at 350 °C, as shown in Figure 10d. This behavior can be explained by the observation that at 350 °C annealing, the creep effect of Cu becomes more pronounced, increasing plastic deformation and thereby increasing Cu protrusion. Therefore, it is predicted that 350 °C is a critical annealing temperature during the heating period. As the annealing temperature increases, the edge of the non-bonding area (point 2) will behave like a crack tip at the SiCN-to-SiCN bonding interface. The crack will initiate from the edge and propagate through the SiCN-to-SiCN bonding interface. 

Next, we investigate the effect of the dishing values on peeling stress as a function of the annealing temperature. Figure 12a,b show the results for the type A and type B structures, respectively. The peeling stress gradually increases with reduction in the dishing value. This is obviously related to Cu protrusion during annealing. The lower dishing value will produce more Cu protrusion. Therefore, a decrease in the dishing value will lead to an increase in the repulsive force between the opposing Cu pads, consequently increasing the peeling stress at the surrounding dielectric interface. 

To assess the risk of crack generation, we compare the calculated peeling stress results with the critical peeling stress of the SiCN-to-SiCN bonding interface (550 MPa). As shown in Figure 12, at the annealing temperatures of 200 °C and 250 °C, the peeling stress is less than the critical peeling stress regardless of the dishing values, indicating a crack-free condition. At 300 °C annealing, the peeling stress is less than the critical peeling stress for dishing values larger than 5 nm. On the other hand, at 350 °C annealing, the calculated peeling stress is larger than the critical peeling stress of SiCN-to-SiCN regardless of the dishing values investigated. The type A structure exhibits slightly higher peeling stress than the type B structure. This suggests that the type A structure is more vulnerable to cracking than the type B structure during the heating period. In summary, it is found that during the heating period, point 2 in region II is the most critical point with high peeling stress. An annealing temperature of 350 °C presents a significant risk of cracking during the heating period. 

#### 3.2.2. Effect of Bonding Conditions During the Cooling Period

During the cooling period, the protruded Cu pads shrink with decreasing temperature, and the elastically deformed Cu pads return to their original shape. The shrinking Cu pads will generate peeling stress at the Cu-to-Cu bonding interfaces. Consequently, these Cu bonding interfaces may undergo debonding, while the dielectric material bonded interface will be subject to compressive stress, which shows different stress behavior compared to the heating period. In this section, we investigate the risk of crack generation during the cooling period. 

Figure 13a shows the von Mises stress distribution for the type A structure, which is subjected to 300 °C annealing with a 5 nm dishing value. The highest von Mises stress appears at the edge of the bottom Cu pad, with values of 788 MPa at point 1 and 1270 MPa at point 2. Figure 13b exhibits the peeling stress (σ_yy_) distribution. High peeling tensile stress appears at the Cu pad interface (point 1), reaching 626 MPa. On the other hand, at point 2 (SiCN-to-SiCN bonding interface), the compressive stress has a value of −1008 MPa, indicating that the probability of cracking at the SiCN-to-SiCN bonding interface is very low. Therefore, during the cooling period, the SiCN-to-Cu bonding interface at the edge of the bottom Cu pad (point 1) is the region most vulnerable to cracking. Similarly, as shown in Figure 14, the type B structure exhibits the highest peeling stress of 467 MPa. Meanwhile, point 2 exhibits compressive stress with a value of −847 MPa. 

Therefore, during the cooling period, the SiCN-to-Cu bonding interface will be the most vulnerable point with regard to crack generation. On the other hand, as mentioned, during the heating period, the SiCN-to-SiCN bonding interface has the greatest vulnerability to crack generation at the bonding interface. 

During the cooling period, besides the risk of cracking at the SiCN-to-Cu bonding interface, there is also a risk of crack formation at the Cu-to-barrier (Ta/TaN) interface in the vertical direction. More specifically, cracks may form due to high tensile stress in the radial direction (x-direction), which is commonly known as radial stress (σ_xx_) [32]. Figure 15a,b display the radial stress distribution in region II at 300 °C annealing and a 5 nm dishing value for the type A and type B structures, respectively. The highest radial stress occurs at the Cu-to-barrier interface, reaching 780 MPa for the type A structure. The radial stress level is high and vertical cracking can also occur at the Cu-to-barrier interface. However, it is thought that the bonding energy between the Cu-to-barrier layer is very high due to the Ta layer enhancing adhesion with Cu. Therefore, the chance of cracking at the Cu-to-barrier interface is expected to be very low. 

The effects of annealing temperature on the peeling stress during the cooling period are illustrated in Figure 16 for the type A structure. In contrast to the heating period, during the cooling period, the peeling stress at the Cu bonding interface decreases with increasing annealing temperature. This can be explained by the fact that a higher annealing temperature will lead to more Cu plastic deformation to cover the dishing volume during the heating period (this is irreversible deformation). Therefore, during the cooling period, the Cu bonding interface will undergo less deformation to maintain the bonding state as the annealing temperature increases, resulting in reduced peeling stress at the Cu bonding interface during the cooling period. Additionally, previous studies by Lin et al. [13] and John Lau [3] reported that a larger bonding area with a higher annealing temperature is also a factor contributing to reducing the peeling stress at the Cu bonding interface during the cooling period [25,51], consistent with our simulation results. At 350 °C annealing, the peeling stress of the Cu bonding interface is significantly lower compared to other annealing temperatures. The significant reduction in the peeling stress is attributed to the substantial creep effect of Cu at 350 °C annealing. 

Finally, the effects of the dishing value for different annealing temperatures on peeling stresses at point 1 during the cooling period are analyzed, as shown in Figure 17. It is evident that during the cooling period, increasing the dishing value will increase the peeling stress at the SiCN-to-Cu bonding interface. This is because a larger dishing value causes the Cu pads to undergo more deformation in order to maintain the bonded state during the cooling period, resulting in an increase in the peeling stress at the Cu bonding interface. Therefore, during the cooling period, less Cu protrusion or a larger dishing value will lead to an increase in the peeling stress. These results are opposite to those recorded during the heating period shown in Figure 12, where more Cu protrusion or a smaller dishing value increase the peeling stress. 

To assess the risk of crack generation, we compare the simulated peeling stress results with the critical peeling stress of the SiCN-to-Cu interface. Previous reports by S. Y. Chang et al. [48,49] indicated that the adhesion strength of the SiCN-to-Cu bonding interface is 1.35–2.21 GPa. As shown in Figure 17, the simulated peeling stresses are lower than the adhesion strength of SiCN-to-Cu regardless of the annealing temperatures and dishing values investigated in this study, indicating that the likelihood of cracking during the cooling process is very low. 

In summary, in order to increase the bonding area and reduce the possibility of crack formation, the optimal process conditions could be as follows: In the case of dishing values in the range of 3 nm to 6 nm, an annealing temperature of around 250 °C should be used. In the case of dishing values greater than 6 nm, the annealing temperature should be 300 °C. The optimal conditions in this study are a dishing value of 3 nm and an annealing temperature of 250 °C. 

## 4. Conclusions

In this study, we investigated the possibility of crack formation caused by misalignment and CMP defects for two common hybrid bonding structures using a numerical simulation. The effects of annealing temperature and dishing values on changes in the non-bonding area and thermomechanical stress were analyzed, and vulnerable regions prone to cracking were also analyzed. 

The non-bonding area in the bonding structure increases with a decreasing annealing temperature and an increasing dishing value. To achieve a non-bonding area of less than 10%, the annealing temperature should be greater than 200 °C. When the annealing temperature reaches 300 °C or higher, the non-bonding area will be less than 10% for all dishing values.During the heating period, the SiCN-to-SiCN bonding interface is the most vulnerable cracking site with the highest peeling stress. The annealing temperature of 350 °C presents a significant risk of cracking. On the other hand, annealing temperature lower than 250 °C will minimize the risk of cracking. The SiCN-to-SiO_2_ bonding interface, which has the lowest adhesion energy and a large CTE mismatch, is another possible cracking site.During the cooling period, the SiCN-to-Cu bonding interface is the most vulnerable point with the highest stress. Decreasing the annealing temperature or increasing the dishing value will increase the peeling stress. The peeling stress is highest at an annealing temperature of 200 °C. However, the simulated peeling stresses are lower than the adhesion strength of the SiCN-to-Cu interface regardless of the annealing temperatures and dishing values, indicating that the probability of cracking during the cooling process is very low. There is also a possibility of crack formation at the Cu-to-barrier interface in the vertical direction. However, the chance of cracking is expected be very low due to the high adhesion strength between the metal films.

## Figures and Tables

**Figure 1 micromachines-15-01332-f001:**
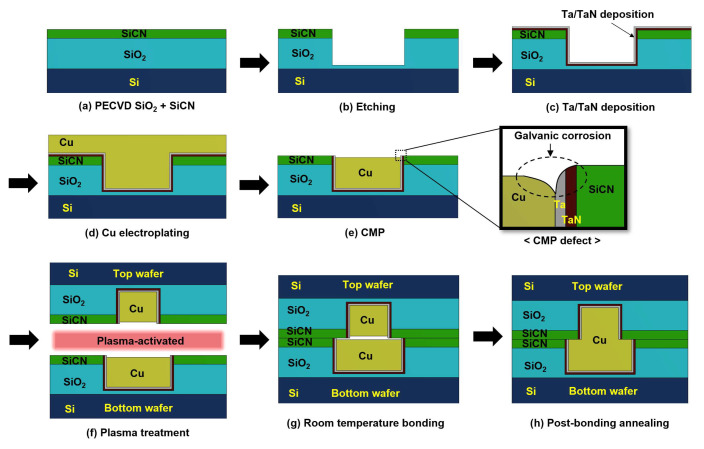
The process flow of hybrid bonding: (**a**) SiO_2_ and SiCN grown on a Si; (**b**) dielectric etching; (**c**) barrier deposition; (**d**) Cu electroplating; (**e**) the CMP process; (**f**) plasma treatment; (**g**) room temperature bonding; (**h**) post-bonding annealing process.

**Figure 2 micromachines-15-01332-f002:**
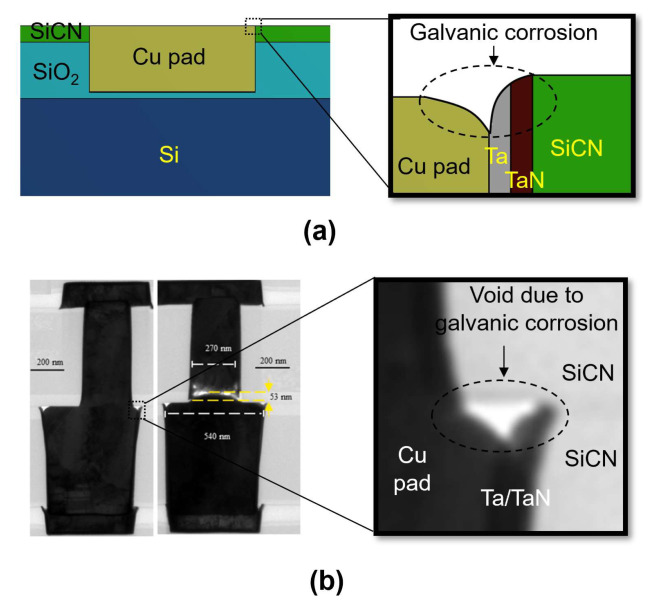
(**a**) Schematics of the effects of galvanic corrosion on the Cu pad and barrier metal caused by the CMP process; (**b**) voids observed at the Cu pad corner after the bonding process due to CMP defects caused by galvanic corrosion (Source: Reprinted with permission from [23] © 2024 Elsevier).

**Figure 3 micromachines-15-01332-f003:**
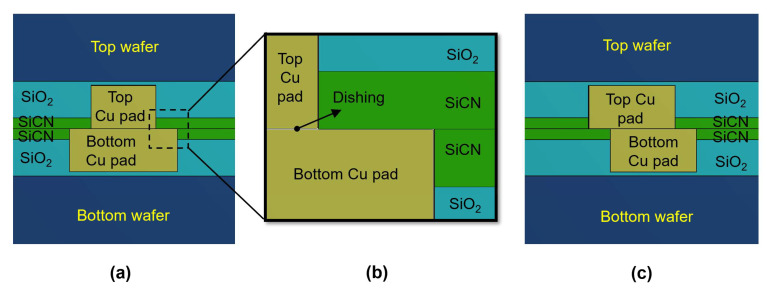
A schematic drawing of hybrid bonding: (**a**) type A structure (the top Cu pad in the top wafer is smaller than the bottom Cu pad in the bottom wafer); (**b**) magnified view of the critical region in the hybrid bonding; (**c**) type B structure (the Cu pads of the top wafer and bottom wafer have the same size).

**Figure 4 micromachines-15-01332-f004:**
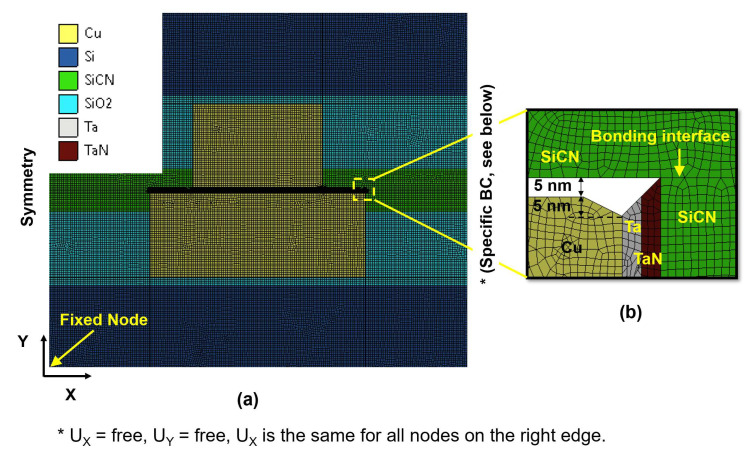
FEA mesh of type A structure for (**a**) entire structure and (**b**) Cu pad corners where galvanic corrosion occurs.

**Figure 5 micromachines-15-01332-f005:**
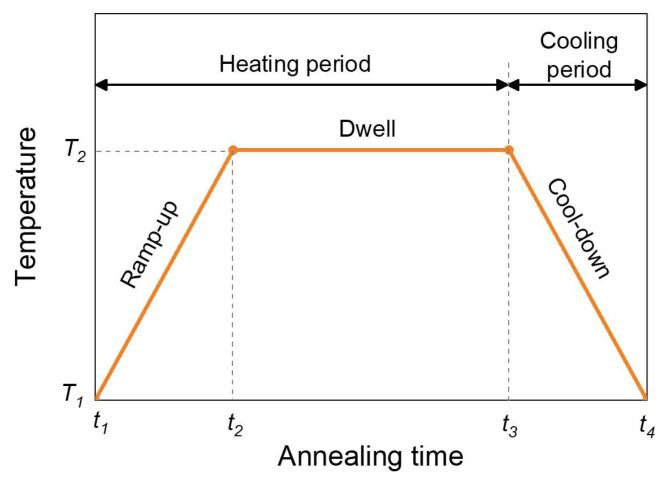
The annealing temperature profile.

**Figure 6 micromachines-15-01332-f006:**
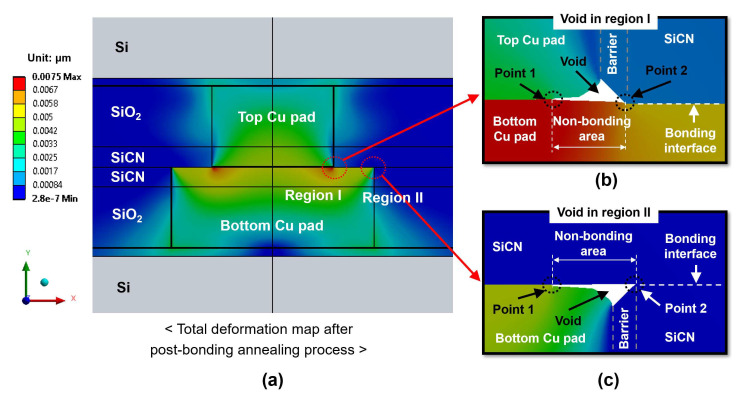
The total deformation map after the annealing process: (**a**) bonding structure; (**b**) void in region I; (**c**) void in region II.

**Figure 7 micromachines-15-01332-f007:**
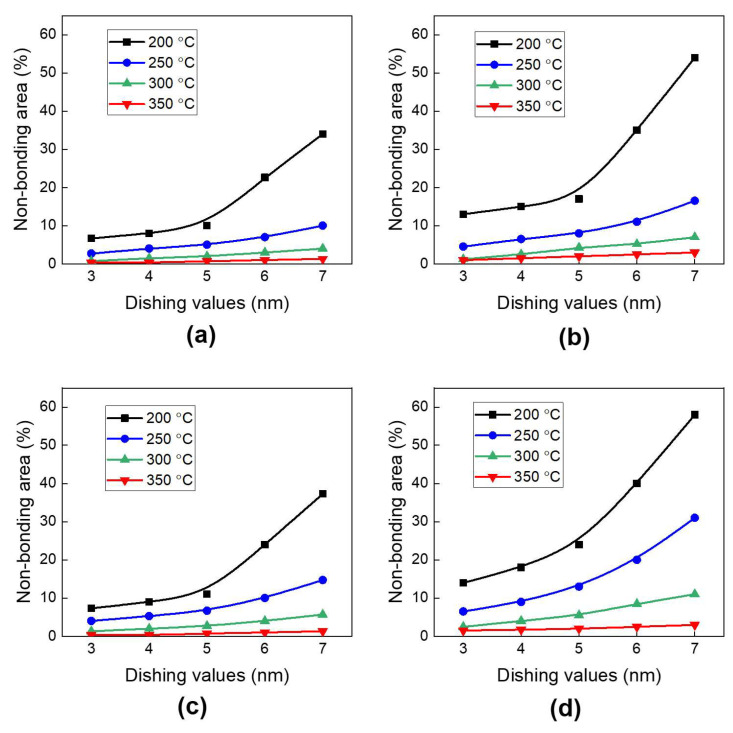
The effects of annealing temperatures and dishing values on the non-bonding area: (**a**) region I of the type A structure; (**b**) region II of the type A structure; (**c**) region I of the type B structure; (**d**) region II of the type B structure.

**Figure 8 micromachines-15-01332-f008:**
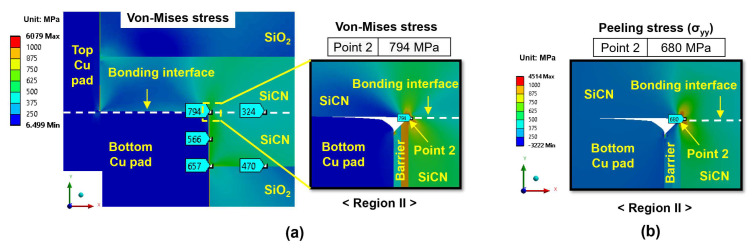
The stress distribution of the type A structure (at 300 °C annealing and a 5 nm dishing value) at the end of the heating period: (**a**) von Mises stress; (**b**) peeling stress (σ_yy_) for region II.

**Figure 9 micromachines-15-01332-f009:**
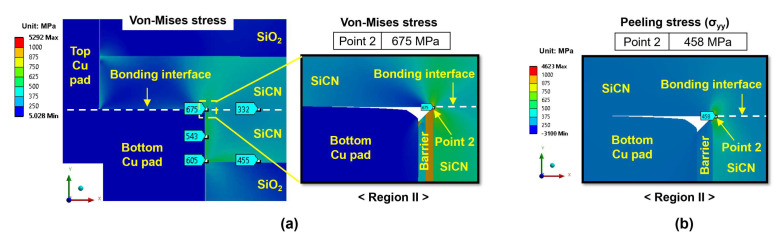
The stress distribution of the type B structure (at 300 °C annealing and a 5 nm dishing value) at the end of the heating period: (**a**) von Mises stress; (**b**) peeling stress (σ_yy_) for region II.

**Figure 10 micromachines-15-01332-f010:**
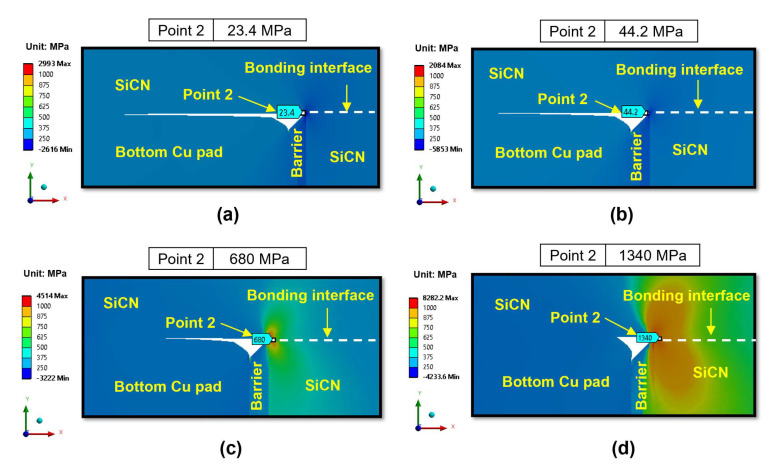
The effects of annealing temperatures on the peeling stress (σ_yy_) of region II at the end of the heating period for the type A structure: (**a**) 200 °C; (**b**) 250 °C; (**c**) 300 °C; (**d**) 350 °C.

**Figure 11 micromachines-15-01332-f011:**
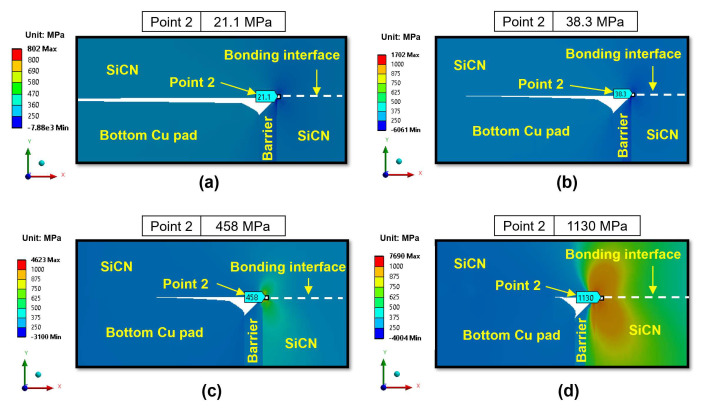
The effects of annealing temperatures on the peeling stress (σ_yy_) of region II at the end of the heating period for the type B structure: (**a**) 200 °C; (**b**) 250 °C; (**c**) 300 °C; (**d**) 350 °C.

**Figure 12 micromachines-15-01332-f012:**
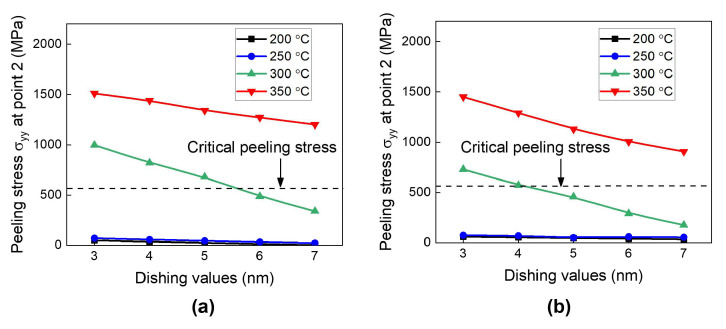
The effect of annealing temperatures and dishing values on the peeling stress (σ_yy_) at the end of the heating period: (**a**) type A structure; (**b**) type B structure.

**Figure 13 micromachines-15-01332-f013:**
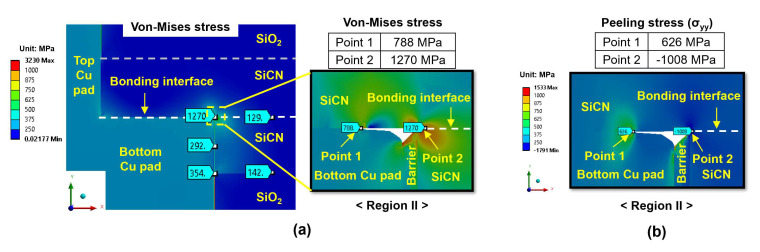
The stress distribution of the type A structure at the end of the cooling period: (**a**) von Mises stress; (**b**) peeling stress (σ_yy_) for region II.

**Figure 14 micromachines-15-01332-f014:**
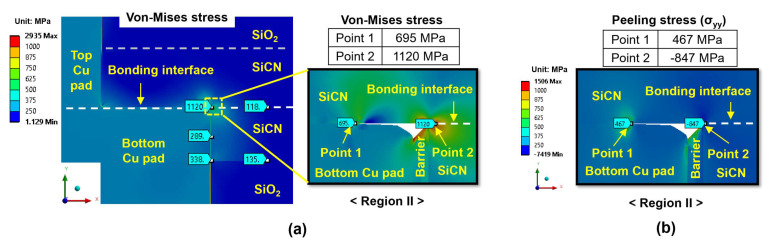
The stress distribution of the type B structure at the end of the cooling period: (**a**) von Mises stress; (**b**) peeling stress (σ_yy_) for region II.

**Figure 15 micromachines-15-01332-f015:**
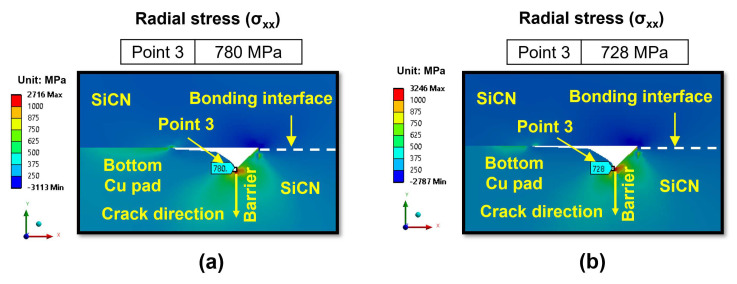
The radial stress (σ_xx_) distribution at the end of the cooling period: (**a**) type A structure; (**b**) type B structure.

**Figure 16 micromachines-15-01332-f016:**
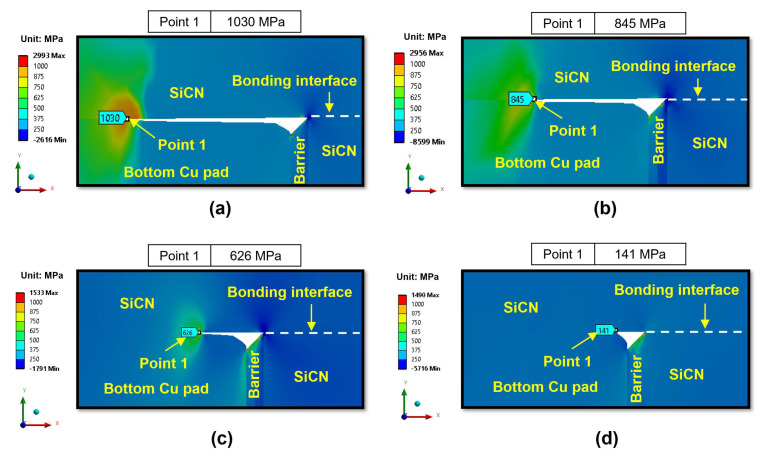
The effect of annealing temperatures on the peeling stress (σ_yy_) of region II at the end of the cooling period for the type A structure: (**a**) 200 °C; (**b**) 250 °C; (**c**) 300 °C; (**d**) 350 °C.

**Figure 17 micromachines-15-01332-f017:**
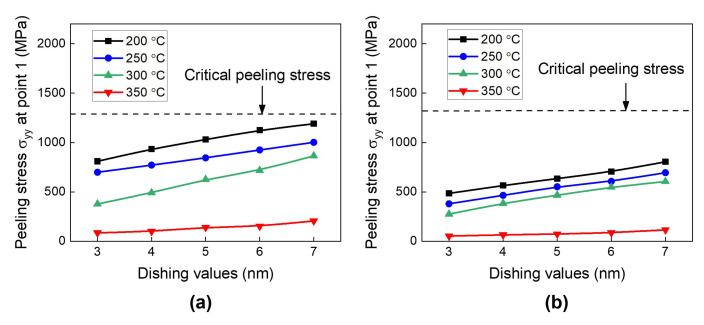
The effect of annealing temperatures and dishing value on peeling stress (σ_yy_) at the end of the cooling period: (**a**) type A structure; (**b**) type B structure.

**Table 1 micromachines-15-01332-t001:** The design parameters considered in this study.

Design Parameters	Values
Dishing values	3~7 nm
Top Cu pad diameter of type A structure	3 μm
Bottom Cu pad diameter of type A structure	5 μm
Cu pads diameter of type B structure	4 μm
Cu pad thickness	2 μm
SiCN thickness	0.5 μm
SiO_2_ thickness	1.7 μm
Si thickness	20 μm

**Table 2 micromachines-15-01332-t002:** The annealing process parameters used in this study.

Annealing Parameters	Values
Initial/end temperature T_1_	25 °C
Annealing temperature T_2_	200, 250, 300, 350 °C
Ramp-up/cool-down duration	0.5 h
Annealing dwell duration	1 h

**Table 3 micromachines-15-01332-t003:** The material properties used in the numerical simulation [37,38,39,40,41,42,43].

Materials	E (GPa)	ʋ	α (ppm/°C)
Cu	91.8	0.34	17.6
Si	131	0.28	2.6
SiO_2_	73	0.17	0.5
SiCN	208	0.31	3
Ta	186	0.34	6.3
TaN	459	0.27	8

E is Young’s modulus, ʋ is Poisson’s ratio, and α is the coefficient of thermal expansion.

## Data Availability

The original contributions presented in the study are included in the article, further inquiries can be directed to the corresponding author.

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
