# Peer review of "Assessment of the Risk of Crack Formation at a Hybrid Bonding Interface Using Numerical Analysis"

_micromachines, 2024, doi:10.3390/mi15111332_

Round 1

Reviewer 1 Report

Comments and Suggestions for Authors

In this manuscript, the authors presented the topic “ Assessment of risk of crack formation in a hybrid bonding interface using numerical analysis. In this work the authors presented the effects of misalignment and CMP defects on the integrity of hybrid bonding using a numerical analysis. They presented a detailed analysis of non-bonding area and crack formation mechanisms during the annealing process and offer detailed guidelines to prevent thermomechanical cracking of hybrid bonding structures. The study is very interesting and provide in-detailed understanding various issues with the hybrid bonding process. There are few comments below the authors need to address.

1.      On what basis different parameters in Table-1 were selected. Please give a detailed reason for different parameters listed in table 1

2.      I understand the complications that arise if the dishing depth is not modeled as flat surface. However, dishing depth should have some roughness. Introducing some roughness to the dishing depth would change the outcome of this study- Please leave a comment.

3.      The authors studied effects of dishing value and annealing temperature on non-bonding area, von-Mises stress, Peeling stress. However, based on all their study, they haven’t suggested an overall optimized process flow for hybrid bonding process. In the summary would it be beneficial if they suggest the optimized dishing value and annealing temperature for achieving peel-free, void free hybrid bonding.

Author Response

Dear Reviewer, 

On behalf of the coauthors, I would like to thanks the reviewer for giving constructive comments on this manuscript. We have sincerely addressed all the comments of the reviewer and revised the manuscript accordingly for improvement. 
Please see the attachment and the corresponding revisions changes in the re-submitted files. 

Thank you again for your valuable feedback.

Reviewer 2 Report

Comments and Suggestions for Authors

The paper provides very careful analysis on the crack mechanism within hybrid bonding technologies. The authors perform the numerical calculations of the mechanical behavior of the components included into bonding structure (Cu, dielectric) as well as of the interfaces between them. As a result of calculation, the authors derive some important dependences of the mechanical strength on the treatment condition and mark most vulnerable parts of the contact for each of the technological operations. The paper might be valuable for improving the contact technology.

However, the motivation part and the basis conditions for the calculation are not clearly presented. This impairs understanding of work and decreases the value of the results and conclusions. I could recommend the paper for the publication only if the authors clarify some of the points below.

1. The first sentence “The increasing demand for fine pitch interconnections and high performance has exceeded Moore's Law for the stringent requirements of the semiconductor industry” – seems very challenging. The authors should either support this claim by the appropriate references.

2. The scheme of the hybrid bonding (page 2) is not a common information. For that reason I believe the authors should describe it in some more details. Maybe this scheme is presented at Fig.1?

3. Continuing the question 2: the Fig.1 and following description of Fig.1 seem unclear and lacking details. What kind of contact is this? How figures (a) and (b) are related?

4. Please provide more details on the following:

- “dry plasma-activated dielectric surfaces” (line 104)

- how and why do “Cu pads on the top and bottom wafers contact each other”? (line 113).

- where is “surrounding dielectric interface” (line 117)

I believe that much more detailed schematic picture illustrating the processes discussed in this section would answer the most of the questions above.

Author Response

(The authors gave the same response as above.)

Reviewer 3 Report

Comments and Suggestions for Authors The manuscript titled “Assessment of risk of crack formation in a hybrid bonding interface using numerical analysis” investigated the effects of misalignment and CMP defects on the integrity of hybrid bonding using a numerical analysis. The possibility of crack formation during the annealing process due to CMP defects caused by galvanic corrosion was investigated. The thermomechanical stress of two common Cu pad structures having misalignment for different annealing temperatures and dishing depths were analyzed and the vulnerable regions prone to cracking were identified as the corners of non-bonding regions. In results, the in-depth insights into the non-bonding area and crack formation mechanisms during the annealing process were provided and detailed guidelines to prevent thermomechanical cracking of hybrid bonding structures were given. After reviewing, the numerical analysis of the manuscript are believed to be helpful for optimize the process of hybrid bonding. The manuscript is suggested to be published in Micromachines after minor revision. Please consider the following comments: a. The authors determine whether cracks has occurred by comparing the stress values obtained in numerical analysis with the adhesive strength of the SiCN-SiCN interface. However, since the crack occurs in the corner of the non-bonding area, the adhesive strength here may not reach the average adhesion strength of the SiCN-SiCN bonding interface. Will this affect the conclusions of the manuscript? b. In Table 2, the duration of heating and cooling is defined as half an hour. Will different durations affect the conclusions of the manuscript? The annealing residence time is defined as one hour. Will different durations have an impact? c. In the conclusion section, it seems misalignment induced no significant drawbacks on increasing non-bonding area or risk of cracking, is it right?

Author Response

(The authors gave the same response as above.)

Round 2

Reviewer 2 Report

Comments and Suggestions for Authors

The authors have clarified all of the issues by providing additional scheme and giving detailed comments. The motivation of the paper looks much better now. I believe the paper can be published in the present form.